

# Habitat suitability and connectivity inform a co-management policy of protected area network for Asian elephants in China

Cheng Huang[1,2], Xueyou Li[1], Laxman Khanal[3] and Xuelong Jiang[1]

[1] Kunming Institute of Zoology, Chinese Academy of Sciences, Kunming, China
[2] Kunming College of Life Sciences, University of Chinese Academy of Sciences, Kunming, China
[3] Central Department of Zoology, Institute of Science and Technology, Tribhuvan University, Kathmandu, Nepal

Corresponding author
Xuelong Jiang,
jiangxl@mail.kiz.ac.cn

## ABSTRACT

Enlarging protected area networks (PANs) is critical to ensure the long-term population viability of Asian elephants (*Elephas maximus*), which are threatened by habitat loss and fragmentation. Strict policies of PAN enlargement that focus on wildlife conservation have failed largely due to difficulties in encouraging stakeholder participation and meeting the elephant habitat requirement. A co-management policy that promotes sustainable resource use, wildlife conservation, and stakeholder participation may have greater feasibility than the strict policies in a developing world. Here, we identified the suitable habitat of elephants using maximum entropy models and examined whether habitat suitability is indirectly associated with local economic development in human-dominated landscapes. We found that (1) the suitable habitat was mainly in areas of forest matrix (50% natural forest cover) with multiple land-use practices rather than relatively intact forest and near communities (mean distance two km) and (2) habitat suitability was negatively associated with local economic development ($r_P = -0.37$, $P = 0.04$). From the standpoint of elephant habitat and its socio-economic background, our results indicate that co-management will be more effective than the currently strict approaches of enlarging PAN. Additionally, our results provide on-ground information for elephant corridor design in southern China.

## INTRODUCTION

Protected area networks (PANs) typically comprise core protected areas (PAs) and corridors that are the cornerstones for ensuring long-term population viability of wildlife by safeguarding contiguous habitat (*Wilson & MacArthur, 1967*; *Bennett & Mulongoy, 2006*; *Geldmann et al., 2013*). Although PAN coverage was markedly increased over the past century with 15% of global land protected in 2018 (https://livereport.protectedplanet.net/chapter-2), some half of PAs were established primarily for preserving natural ecosystem similar to PAs of IUCN categories I–IV, that is, nature reserve (NR), wilderness area, national park, natural monument, and habitat/species management area

(*McDonald & Boucher, 2011*), where human activities are strictly restricted. These strict policies generate three concerns from conservation fields. First, the habitat suitability of some species and taxa in strict PAs might be decreased over time due to lack of landscape heterogeneity (*Wharton, 1968*; *Mudappa et al., 2007*; *Evans, Asner & Goossens, 2018*). Second, PA-oriented efforts lead to increased isolation of PAs and wide-ranged species (*DeFries et al., 2005*; *Laurance et al., 2012*) because primary and secondary vegetation in human-dominated landscapes are continually eroded (*Joppa & Pfaff, 2009*; *Acharya et al., 2017*; *Evans, Asner & Goossens, 2018*). Third, encouraging local stakeholder participation is difficult especially in developing countries because the establishment of strict PAs and economic development are commonly regarded as competing issues by local stakeholders (*Bennett & Mulongoy, 2006*; *McDonald & Boucher, 2011*). In this context, a co-management policy that promotes sustainable resource use, wildlife conservation, and stakeholder participation potentially provides a more feasible mean for PAN enlargement for some species or taxa in human-dominated landscapes (*Zhang, Ma & Feng, 2006*; *Goswami et al., 2014*; *Evans, Asner & Goossens, 2018*).

Several global biodiversity hotspots are found in south and southeast Asia (*Myers et al., 2000*), where wildlife is threatened by human activities (e.g., agriculture and infrastructure) (*Ceballos & Ehrlich, 2002*; *Edwards et al., 2010*; *Hansen et al., 2013*; *Clements et al., 2014*). Large animals are particularly affected because of their wide range (*Ceballos & Ehrlich, 2002*; *Robert, Wanlop & Naret, 2006*) and negative interactions with villagers (*Acharya et al., 2017*; *AsERSM, 2017*). Although Asian elephants (*Elephas maximus*) are endangered species and are important in ecosystem function (e.g., seed dispersal and nutrient recycling), culture, and fundraising for wildlife conservation (*Campos-Arceiz et al., 2008*; *Ritchie & Johnson, 2009*; *Verissimo, Macmillan & Smith, 2011*), only 29% of their distribution range is legally protected in 13 countries (*Hedges, Fisher & Rose, 2008*), and most is in human-dominated landscapes (*Jathanna et al., 2015*; *Calabrese et al., 2017*). Enlarging PANs was suggested as a priority for their conservation (*AsERSM, 2017*). However, today, economic development is the top priority in many regions, and thus attempts to expand PANs with the strict policies is likely to fail socially (*Bennett & Mulongoy, 2006*; *Zhang, Ma & Feng, 2006*; *Evans, Asner & Goossens, 2018*).

Strict PAN might also be failed to meet the elephant habitat requirement. Asian elephants are habitat generalists that use primary and secondary forests, scrubland, grassland, and farmland (*Choudhury et al., 2008*), and their resource-use and safety strategies are context-dependent. For instance, in China, the Cangyuan population (20–23 individuals) tend to stay within an area of ~33 km$^2$ in an NR (*Liu et al., 2016*); the Mengla–Shangyong population (88–98 individuals) is located within two subdivisions (1,239 km$^2$) of an NR and its periphery (*Chen et al., 2013b*); the Menghai–Lancang population (15 individuals) and most of the Xishuangbanna–Pu'er population (98–109 individuals) frequently use human-dominated landscapes (Fig. 1). Despite these differences, there is mounting evidence that Asian elephants are forest-edge specialists at the fine spatial scale (*Sitompul et al., 2013*; *Wadey et al., 2018*). However, strict PAN substantially reduces human resource use and fire incidence (*Nelson & Chomitz, 2011*), resulting in intact closed

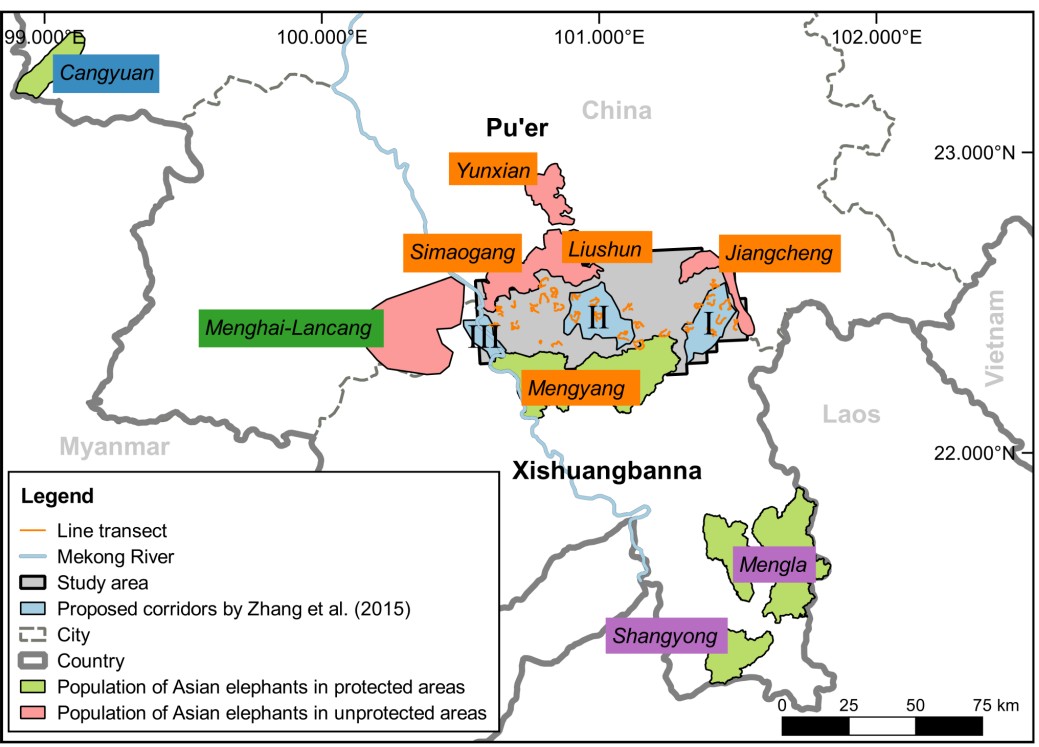

**Figure 1 The study area and distribution range of Asian elephants in China.** The populations are represented by the tags of orange (Xishuangbanna–Pu'er population), blue (Cangyuan population), green (Menghai–Lancang population), and purple (Mengla–Shangyong population).

forests, which are less suitable for elephants than moderately disturbed forests (*Sitompul et al., 2013*; *Evans, Asner & Goossens, 2018*; *Wadey et al., 2018*).

On the other hand, elephants cause extensive damage to villages by raiding crops, damaging property, and even killing people (*Gubbi, 2012*; *Chen et al., 2016*). Areas with severe damage or frequently used by Asian elephants are typified by hilly terrain with traditional farming practices and relatively far from major roads (*Wilson et al., 2013*; *Chen et al., 2016*). Villages in these areas are generally less developed economically than villages located in areas with flat terrain and large cash-crop plantations near major roads. Thus, alternative supports to these villages are necessary to offset elephant-caused losses and encourage villager participation in enlarging PAN for elephants.

Here, we propose that a co-management policy that integrates sustainable resource use, wildlife conservation, and stakeholder participation is more feasible than the currently strict policies that only focus on wildlife conservation. This proposition will be supported by two key pieces of evidence. First, areas of relatively intact forest are less suitable for elephants than forest matrix with multiple land-use practices. Second, habitat suitability is negatively associated with local economic development; namely, areas of poorer villages provide more suitable habitat than areas of relatively wealthy villages. Our study provides useful information to guide conservation policy to improve PAN enlargement and corridor design for elephant conservation.

## MATERIALS AND METHODS

### Field permit

Field studies were conducted under the permission from the Yunnan Forestry and Grassland Administration.

### Study area

This study was conducted within the range of the Xishuangbanna–Pu'er population in Xishuangbanna and Pu'er, Yunnan, southwest China, bordering Vietnam and Laos (Fig. 1). This population comprises five subpopulations: Liushun, Yunxian, Simaogang, Jiangcheng, and Mengyang (Fig. 1). The region ranges from 495 to 1,851 m above sea level, with an annual mean temperature of 21 °C and annual precipitation of ~1,500 mm (*Liu et al., 2018*). Natural forests (mainly subtropical evergreen broad-leaved forest) are fragmented by production forests (e.g., *Pinus kesiya* and *Eucalyptus* spp.), cash-crop plantations (e.g., rubber, coffee, and tea), and traditional farmlands (e.g., corn, rice, and sugarcane) (*Chen et al., 2010*). Three corridors (I, II, and III) were proposed by *Zhang et al. (2015)* to connect the (a) Menghai–Lancang and Xishuangbanna–Pu'er population and (b) subpopulations of the Xishuangbanna–Pu'er population (Fig. 1). However, the Jinghong hydro-power dam raised the water level of the Mekong River, isolating the Menghai–Lancang population from the Xishuangbanna–Pu'er since 2005 (*Chen et al., 2010*). The study area includes 32 villages, each of which comprises several communities (251 in total). A town is the social center of villages and usually comprises several adjacent communities. The primary industries are agriculture and agroforestry (*Chen et al., 2010*).

### Data collection

In the confirmed range, we collected data on elephant presence and land-cover along 91 line transects (307 km) from December 2016 to March 2017, with the assistance of forest rangers. These line transects were designed to traverse all land-cover types (Figs. 1 and 2). Dung piles and footprints within a 20 m width of the line transects were recorded, with intervals of at least 200 m (Dataset S1). Land-cover was categorized into seven types: that is, natural forest, pine plantation (i.e., *P. kesiya*), cash-crop plantation, shrubland, traditional farmland, infrastructure site (e.g., settlements and roads), and water body (i.e., rivers, reservoirs, and ponds) (*Chen et al., 2010*).

We treated the per-capita annual income of village as a proxy for economic development, with higher incomes representing higher levels of economic development. The data was collected from the Digital Village of Yunnan (http://ynszxc.gov.cn/S1/).

### Data analysis

The analysis included five steps. First, environmental variables were selected for habitat suitability models. Second, a land-cover map was developed from remote-sensing images. Third, maximum entropy models (MaxEnt) were used to identify suitable habitat of elephants. Fourth, the elephant pathways were simulated by least-cost and circuit models.

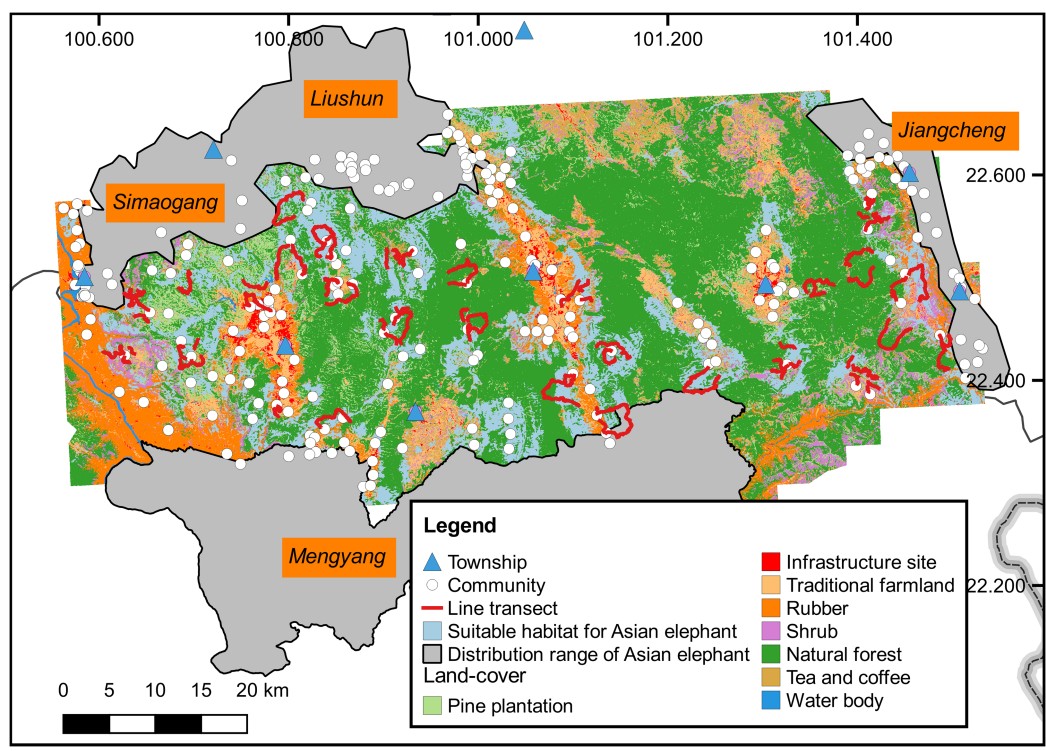

**Figure 2 Habitat suitability map for Asian elephants in the study area.** The suitable habitat of Asian elephants was mainly distributed in the areas of forest matrix with multiple land-use, away from towns and near community settlements.

Fifth, the potential negative association between habitat suitability and level of economic development was examined by Pearson's correlation.

## Environmental variables

Asian elephants frequently occur in areas of low altitude, flat terrain, and low human disturbance and feed on natural foods or crops near forest edge (*Jathanna et al., 2015*; *Lin et al., 2015*; *Liu et al., 2016*). Hence, we selected 13 environmental variables in three categories for habitat suitability models (Table 1): that is, geographic and topographic (altitude and terrain roughness index), land-cover (distance to, edge density of, and percentage of natural forest, pine plantation, and traditional farmland), and human disturbance (distance to town and distance to community).

## Land-cover classification

We used Landsat 8 OLI_TIRS images (30 m resolution from the Data Cloud of CAS, http://www.csdb.cn/) to develop a land-cover map. We added ancillary layers to improve classification accuracy, including ASTER GDEM grids (the Data Cloud of CAS), slope and its texture, and normalized difference vegetation index and its texture (*Wegmann, Leutner & Dech, 2016*). We performed a supervised classification using the random forest algorithm with 25% of land-cover points left to validate the classification (*Leutner & Horning, 2017*).

**Table 1 Environmental variables selected in habitat suitability models for Asian elephants.**

| Category | Variable | Data and calculation |
|---|---|---|
| Geographic and topographic | Altitude | ASTER GDEM |
| | Terrain roughness index | Calculated from ASTER GDEM in R |
| Land-cover | Distance to natural forest<br>Distance to pine plantation<br>Distance to traditional farmland | Calculated by "distance" function in R |
| | Percentage of natural forest<br>Percentage of pine plantation<br>Percentage of traditional framland | Calculated in Fragstats by 1.5 km radius from land-cover map |
| | Edge density of natural forest<br>Edge density of pine plantation<br>Edge density of traditional framland | Calculated in Fragstats by 1.5 km radius from land-cover map |
| Human disturbance | Distance to town<br>Distance to community | Calculated by "distance" function in R |

## MaxEnt modeling

For habitat suitability models with presence-only data, MaxEnt outperforms other existing approaches (*Ferrier et al., 2006*; *Phillips, Anderson & Schapire, 2006*). MaxEnt contrasts environment of wildlife presences against the available background (*Elith et al., 2011*). Here, the background was represented by 10,000 points randomly generated in buffer zones of average home range size (113 km$^2$) around the presence points (Dataset S2) (*Fernando et al., 2008*; *Amirkhiz et al., 2018*).

To identify important environmental variables describing habitat suitability and build a model with high accuracy, we performed an optimized selection of variables and MaxEnt features and β multiplier based on Akaike information criteria (AIC) following the workflow of *Amirkhiz et al. (2018)*. First, each model included variables that were not highly correlated ($|r| \leq 0.7$) and that had a model contribution >5% and then step-wise optimized the β multiplier from zero to 15 at an increment of 0.5. Second, as MaxEnt calculates five models for each variable, known as features (i.e., linear (L), quadratic (Q), product (P), threshold (T), and hinge (H)) (*Phillips et al., 2017*), we selected feature sets by the lowest AIC among "L," "H," "LQ," "LQT," "LP," "HP," "LQP," and "LQTP," then used the optimized model to predict a habitat suitability map. The prediction was evaluated by threshold-independent (i.e., area under the curve of the receiver operating characteristic plot, AUC) and threshold-dependent omission rate. Third, a 10% training presence threshold was used for delineating the suitable from unsuitable habitat (*Escalante et al., 2013*, *Hughes, 2017*), after which we summarized the characteristics of the suitable habitat. The modeling was performed in R with MaxentVariableSelection and ENMeval package (*R Development Core Team, 2013*; *Muscarella et al., 2014*; *Jueterbock et al., 2016*).

## Pathway mapping

Least-cost and circuit models are two widely used approaches for animal corridor design (*Ruiz-González et al., 2014*; *Wang et al., 2014*). We simulated the elephant pathways by least-cost and circuit models using Linkage Mapper and Circuitscape software

(*McRae & Shah, 2009*; *Wang et al., 2014*; *Mcrae et al., 2008*), in which the length and resistance of the least-cost paths were calculated. The resistance surface was calculated by one minus the habitat suitability layer. As we focused on mapping pathways around the previously-proposed corridors (I, II, and III) by *Zhang et al. (2015)*, the least-cost model was constructed with three core ranges, that is, Mengyang, Liushun and Simaogang, and Jiangcheng (Fig. 1). All presence points were used to produce a connectivity map for the entire study area by circuit model.

## Association between habitat suitability and level of economic development

In the study area, economic development of a village is a consequence of its altitude, terrain, and land-use practices and thus may be indirectly associated with habitat suitability of the elephants. The habitat suitability of a village was calculated by averaging that of communities, which were extracted from the habitat suitability map by community locations. We used Pearson's correlation to examine the direction and significance of the association between habitat suitability and level of economic development.

## RESULTS

We collected 245 presence points of Asian elephants. The overall accuracy of the land-cover map was 0.91. The model with the lowest AIC had a $\beta$ multiplier = 1; LQTP features; and eight variables, including terrain roughness index, distance to town, distance to community, distance to natural forests, distance to traditional farmlands, percentage of natural forest, percentage of pine plantation, and percentage of traditional farmland. The percentage of natural forests (23%), distance to town (23%), and distance to community (16%) were among the strongest predictors of the elephant presence.

In general, the optimized model accurately discriminated the presence points from the background environment (mean AUC = 0.86). The low AUC difference (0.05) suggested that the model did not over-fit the presence points. Threshold-dependent measures indicated that the model had low over-fitting and high discriminatory ability at 10% omission rate (0.20) and lowest presence threshold (<0.001). The threshold value of the suitable habitat was 0.28. In our study, the suitable habitat of Asian elephants was mainly found in areas of forest matrix (50% natural forest cover) with multiple land-use practices rather than relatively intact forest, away from towns (mean distance 10 km), near communities (mean distance 2 km), and with flat terrain (mean terrain roughness index 4.83) (Fig. 2).

The least-cost model (Fig. 3) demonstrated that the shortest pathway is #3 (29 km) and the longest pathway is #2 (47 km), while pathway #1 had the lowest habitat resistance. The connectivity map of the study area supported pathway #1 as a potential corridor to connect the Mengyang and Jiangcheng subpopulation (Fig. 3). Additionally, the connectivity map showed that the area of the white rectangle on Fig. 3 is important in connecting the subpopulations of Mengyang to Liushun and Simaogang because of its location and relatively high habitat connectivity.
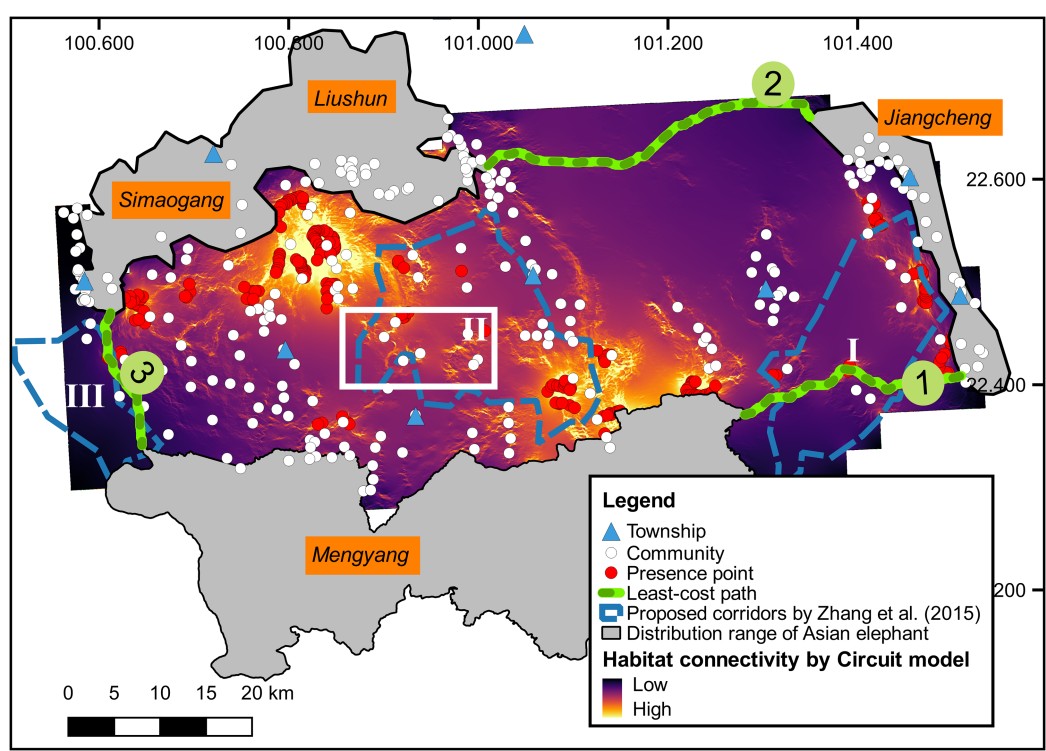

**Figure 3 Habitat connectivity for Asian elephants calculated by the circuit model and the least-cost path in the study area.** The area of the white triangle is located among the subpopulations of Mengyang, Liushun, and Simaogang.

Habitat suitability of elephants was negatively associated with level of economic development ($r_P = -0.37$, $P = 0.04$). Thus, with current land-use practices, areas of poorer villages provided more suitable habitat than areas of relatively wealthy villages.

## DISCUSSION

For elephants, habitat selection reflects a trade-off between resource use and mortality risk (*Munshi-South et al., 2008*; *Basille et al., 2009*). Here, natural forest was the strongest variable influencing the presence of Asian elephants (as elsewhere, *Liu et al., 2016*) and indicates the substantial role natural forest has for the elephants with respect to food, refuge and thermoregulation (*Kumar, Mudappa & Raman, 2010*; *Goswami et al., 2014*; *Evans, Asner & Goossens, 2018*). In particular, forest matrix (50% natural forest cover) with multiple land-use practices are more suitable for the elephants than relatively intact forest in human-dominated landscapes (*Sitompul et al., 2013*; *Evans, Asner & Goossens, 2018*; *Wadey et al., 2018*). Forest edges provide better light conditions for *Ficus* spp. and grasses that are primary natural foods of elephants (*Chen et al., 2006*; *Sitompul et al., 2013*; *Wadey et al., 2018*). Also, crops in the forest matrix are attractive to the elephants, with 68% of feeding sites in such areas during the rainy season (*Zhang et al., 2003*). On the other hand, elephants suffer mortality at the hands of humans, both directly and indirectly, from ditch, electrocution, and retaliatory killing (*Chen et al., 2013a*; *Palei et al., 2014*; *AsERSM, 2017*). As a consequence, Asian elephants are less likely to occur near

towns with dense human population, infrastructure, and plantation (Fig. 2). Although we focused on habitat suitability patterns of the elephants in human-dominated landscapes, similar patterns can be found in NRs and their peripheries. For example, the Mengla–Shangyong population mostly inhabits the buffer and experimental zones of an NR and its peripheries with moderately disturbed landscapes (Fig. 1) (H. Yang, 2017, personal communication).

Based on the quantitative analysis, efforts on establishing corridors for the elephants should be concentrated on the predicted pathways and areas of high connectivity. With the greatest length and largest movement resistance, pathway #2 was rarely used by the elephants (based on long-term monitoring of *Chen et al. (2010)* and *Zhang et al. (2015)*). Despite having the shortest length, the resistance of pathway #3 was only slightly less than that of pathway #2 and traversed tracts of rubber plantations (Fig. 2), where stakeholders are unlikely to restore contiguous natural habitat for the elephants. Pathway #1 was the most consistent with the connectivity map calculated by the circuit model and had the lowest resistance. Thus, pathway #1 should be allocated greater conservation priority than pathway #2 and #3. Also, efforts are needed to protect the connective habitat of the area with the white rectangle on Fig. 3. Our study provides more precise information for elephant corridor design than *Zhang et al. (2015)*.

Habitat suitability of Asian elephants is affected by many factors. Our study is limited by our reliance on presence-only data and variables extracted from remote sensing images to determine the habitat suitability, from which the resistance layer was generated for simulating pathways. Incorporating movement data of elephants recorded by telemetry techniques and on-ground variables (e.g., food abundance and forest structure) could improve habitat suitability models and provide straightforward movement trajectories for corridor design.

In China, PANs include NRs (~15% of the national territory), world natural and cultural heritage sites, scenic zones, wetland parks, forest parks, geological parks, and water conservancy scenic locations (*Cao, Peng & Liu, 2015*). While most NRs are managed as socially exclusive landscapes (*Zhang, Ma & Feng, 2006*; *Cao, Peng & Liu, 2015*), including the Xishuangbanna National Nature Reserve (soft green area in Fig. 1), Asian elephants need forest matrix with open lands and are flexible to human disturbance. Conservation policies allowing considerable interventions in NRs could enlarge elephant habitat without great loss of biodiversity. For example, selectively logged forests appear to maintain ~90% of the original biodiversity compared to primary forest (*Berry et al., 2010*; *Brodie et al., 2014*), and retention forestry, whereby a proportion of original vegetation is left unlogged, further reduces the negative impacts on biodiversity (*Gaveau et al., 2013*; *Fedrowitz et al., 2014*). Among NRs, efforts should be paid to protect community-owned forests, which represent a major proportion of natural forests and are critical for elephants (*Kumar, Mudappa & Raman, 2010*; *Evans, Asner & Goossens, 2018*) and other wildlife (*Rodrigues et al., 2017*; *Rodrigues & Chiarello, 2018*). Meanwhile, integrating traditional farmlands into PANs can fulfill human needs and encourage the participation of villagers. Generally, the less-developed villages are more suitable to the elephants than are the more-developed villages. Thus, supporting sustainable

economic development and reducing elephant-caused losses are needed to encourage human-elephant coexistence, and may include developing ecotourism, encouraging wildlife-friendly products, and compensating the losses (*Mishra et al., 2003*; *Chen et al., 2013b*; *Huang et al., 2018*).

## CONCLUSIONS

Asian elephants are globally threatened by habitat fragmentation and loss. Thus, enlarging PANs is the current priority for elephant conservation (*AsERSM, 2017*). Using presence data from an on-ground survey in human-dominated landscapes combined with habitat suitability models, we found that: (1) suitable habitat of the elephants was mainly in areas of forest matrix with multiple land-use practices rather than relatively intact forests and near communities; and (2) habitat suitability and level of economic development had an inverse correlation. From the standpoint of the elephant habitat and its socio-economic background, our results suggest that a co-management policy would be more feasible than the currently strict policies for enlarging PANs. Such a policy would also be suitable for other areas with similar land-cover practices and socio-economic contexts, such as northeastern India and northern Laos (*Kumar, Mudappa & Raman, 2010*; *Wilson et al., 2013*; *AsERSM, 2017*).

## ACKNOWLEDGEMENTS

We are grateful to the Pu'er Forestry Bureau, Xishuangbanna National Nature Reserve, and Jinghong Forestry Bureau for field support. We thank Hongpei Yang and Wei Cha for sharing their experiences, and Zhonghua Li, Li He, and Dan Yan for their assistance in the field.

### Funding

This study was supported by the Strategic Priority Research Program of the Chinese Academy of Sciences (Grant No. XDA23080501) and Wildlife Conservation Programme of Yunnan, China. The funders had no role in study design, data collection and analysis, decision to publish, or preparation of the manuscript.

### Grant Disclosures

The following grant information was disclosed by the authors:
Strategic Priority Research Program of the Chinese Academy of Sciences: XDA23080501.
Wildlife Conservation Programme of Yunnan, China.

### Competing Interests

The authors declare that they have no competing interests.

### Author Contributions

- Cheng Huang conceived and designed the experiments, performed the experiments, analyzed the data, contributed reagents/materials/analysis tools,
prepared figures and/or tables, authored or reviewed drafts of the paper, approved the final draft.

- Xueyou Li conceived and designed the experiments, analyzed the data, contributed reagents/materials/analysis tools, authored or reviewed drafts of the paper, approved the final draft.
- Laxman Khanal authored or reviewed drafts of the paper, approved the final draft.
- Xuelong Jiang conceived and designed the experiments, authored or reviewed drafts of the paper, approved the final draft.

## Field Study Permissions

The following information was supplied relating to field study approvals (i.e., approving body and any reference numbers):

Field works were approved by Yunnan Forestry and Grassland Administration.

## Data Availability

Raw data are available in the Supplemental Files.

## Supplemental Information

Supplemental information for this article can be found online at http://dx.doi.org/10.7717/peerj.6791#supplemental-information.

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
