# Peer review of "Habitat suitability and connectivity inform a co-management policy of protected area network for Asian elephants in China"

_PeerJ, doi:10.7717/peerj.6791_

## Round 0.1 · original submission · Major Revisions

Dear Dr. Cheng,

I have read the manuscript and decided that it is not yet acceptable to be sent to reviewers. There are several problems with the manuscript, although I feel it has potential if these problems are addressed. First, the English language has many problems - often incomplete sentences, incorrect grammar and so on. I comment on many of them in the annotated manuscript that includes my comments. Second, there are several scientific and methodological problems with the structure of the research as described in the manuscript. The first and foremost is that you seem to test that elephants prefer mixed habitat to forest interior, yet that is well-known elephant ecology and so you are not truly testing that.

Several terms are used in the manuscript without adequate context or definition, as you will see in my comments. I attempted to illustrate several kinds of issues with writing and language, but I did not exhaustively review all details. You should use my comments as a guide and look for similar errors in the remaining parts of the manuscript.

To reiterate, I felt that this version of your manuscript should not be sent to reviewers because I first identified many problems. I recommended major revisions in order to keep the possibility open to be resubmitted, and to then enter formal peer-review.

.

---

## Round 0.2 · Minor Revisions

Both reviewers agreed that there are useful data and ideas to be presented here. Several of their comments are about writing clearly in English, so please pay particular attention to their points. Because of the details of MaxEnt and the issues it has with absences and pseudo-absences, and because you are talking about habitat "selection" (and one reviewer made very pertinent comments on what habitat selection and preferences really are), please make sure you clearly define variables, and how you define habitat choice and selection. One of the reviewers provided an annotated copy of your manuscript, so please address the issues contained therein.

Reviewer 1 ·

Basic reporting

No comment.

Experimental design

No experiment was conducted, as this study is purely descriptive. However, the methods are generally adequate and appropriate.

Validity of the findings

The results support the conclusions, so long as semantic issues are rectified (see below).

Additional comments

The authors claim to measure habitat suitability and habitat selection, which is actually somewhat misleading. Habitat suitability typically refers to how fitnesses of individuals vary across habitats, while habitat selection infers an active choice by individuals and can only be determined experimentally. The authors are actually measuring habitat associations, which are simply statistical relationships between individual occurrences and various habitat features. I see this as more of a semantic issue that is easily rectified rather than a real problem with the methods. For instance, on line 223, it is stated that “elephants prefer forest matrix habitats…”. Just because there was evidence of elephants in those habitats does not necessarily imply that those habitats are preferred and would be most suitable (i.e., where fitnesses would be highest), especially given the great mobility and movement distances of elephants. They may simply be moving through in an attempt to find more appropriate habitat. A more accurate indicator of “preferences” would be the amount of time spent in each habitat and what they are doing in those habitats. The authors allude to that beginning on line 253, but it could perhaps be clarified. I suggest rewording parts of the manuscript, being careful to avoid using “habitat suitability” and “habitat selection” (e.g., on line 234 – that is simply associations) where that wording is misleading. “Habitat selection” on line 253 is fine because the authors are referring to a concept rather than what they actually measured.
Additional minor comments follow:
Line 57: “South and Southeast Asia are global hotspots): replace “are” with “contain”.
Line 64: “umbrella species” could be clarified.
Line 94: “co-management policy” and “a strict policy” are vague in this context.
Line 107: a reference should be cited for the meteorological data.
Line 124: “recorded with at least 200 m intervals” is confusing.
Line 143: I do not see that there is an explanation of how “town” and “community settlement” differ. Perhaps I am overlooking it.
The headings “Independent variables”, “Land-cover classification”, “MaxEnt modeling”, “Association between habitat suitability and level of economic development”, and “Pathway mapping” are all subsections of “Data analysis” rather than sections that are distinct from that section.
Line 268: “community-own” should be “community-owned”.

·

Basic reporting

Generally, the article is concise and uses professional English. The goals, methods, results, discussion and conclusions are well-framed. There are numerous cases, annotated on the manuscript, where a sentence or phrase could be revised to make the authors' ideas clearer. Most of the suggested changes are minor in nature.

Experimental design

The study relies on a variety of sophisticated, and appropriate, techniques.

Validity of the findings

The data appear sound, and the complex techniques appear to have been used properly.

Additional comments

1. Given the flexibility with which the parameters of the MaxEnt model can be chosen, it might be helpful to provide more detail on how the parameters were chosen.
2. The only large error I saw was in the Discussion section, Lines 238-245. The numbers identifying the connecting paths in the text do not correspond to the labels on Figure 3 or to the description of the pathways on Lines 210-215. The discussion should be modified so that the comments refer accurately to Figure 3 and the comments in the Results and Discussion sections are consistent.
3. Several of the "Altitude" entries in the Presence Raw data show as "######", probably because the cells were designated as text. Clicking on the cell shows a number in the box just above the spreadsheet. It might also be useful to indicate the units for the variables listed in the spreadsheets.

---

## Round 0.3 · Minor Revisions

I am including an annotated copy of your manuscript to indicate where I think you can improve the quality of the English in a variety of ways. The manuscript was clearly improved with this last round of corrections where you satisfied the reviewers suggestions, but, as is often the case when writing in a non-native language, additional corrections arise. I took care in commenting on the Abstract at the very beginning, the Introduction and the Discussion. Please pay very close attention to understand what is being improved by my changes, and please look for other, similar, constructions in the rest of the manuscript.

---

## Round 0.4 · accepted · Accept

Your text has improved with each step, which is always good to see. Still, I am sending a copy of your manuscript with a few (very few) points that you might take a look at while in production. Congratulations.

#